# A Single Cell but Many Different Transcripts: A Journey into the World of Long Non-Coding RNAs

**DOI:** 10.3390/ijms21010302

**Published:** 2020-01-01

**Authors:** Enrico Alessio, Raphael Severino Bonadio, Lisa Buson, Francesco Chemello, Stefano Cagnin

**Affiliations:** 1Department of Biology, University of Padova, 35131 Padova, Italy; alessio.enrico88@gmail.com (E.A.); raphaelbonadio@gmail.com (R.S.B.); , francesco.chemello@gmail.com (F.C.); 2CRIBI Biotechnology Center, University of Padova, 35131 Padova, Italy; 3CIR-Myo Myology Center, University of Padova, 35131 Padova, Italy

**Keywords:** single-cell, single-cell sequencing, single-cell expression, non-coding RNAs, long non-coding RNAs, lncRNAs, lncRNA database, single-cell database

## Abstract

In late 2012 it was evidenced that most of the human genome is transcribed but only a small percentage of the transcripts are translated. This observation supported the importance of non-coding RNAs and it was confirmed in several organisms. The most abundant non-translated transcripts are long non-coding RNAs (lncRNAs). In contrast to protein-coding RNAs, they show a more cell-specific expression. To understand the function of lncRNAs, it is fundamental to investigate in which cells they are preferentially expressed and to detect their subcellular localization. Recent improvements of techniques that localize single RNA molecules in tissues like single-cell RNA sequencing and fluorescence amplification methods have given a considerable boost in the knowledge of the lncRNA functions. In recent years, single-cell transcription variability was associated with non-coding RNA expression, revealing this class of RNAs as important transcripts in the cell lineage specification. The purpose of this review is to collect updated information about lncRNA classification and new findings on their function derived from single-cell analysis. We also retained useful for all researchers to describe the methods available for single-cell analysis and the databases collecting single-cell and lncRNA data. Tables are included to schematize, describe, and compare exposed concepts.

## 1. Introduction

The central dogma of molecular biology explains the flow of the information for the synthesis of proteins. It starts from DNA transcription to synthesize the mRNA and concludes with protein synthesis through mRNA translation [1]. In the past, RNA was considered to be only an intermediate between DNA and proteins. However, with the improvements in DNA/RNA sequencing technologies, it has been revealed that RNA is more than a simple go-between molecule. The international ENCODE project has established that 75% of the human genome is transcribed into RNAs but only 2% of these transcripts are translated into proteins [2]. This evidence indicates that the information in 98% of transcribed DNA is not used to synthesize proteins. The huge energetic cost required for such massive transcription of the genome cannot be simply explained by random association of RNA polymerases to DNA: It should be required for important roles that cannot be fulfilled by DNA and proteins [3]. In fact, given the genome size and the number of coding genes found in complex eukaryotes (about 20,400 in *H. sapiens* and *M. musculus*, according to the Ensembl and Mouse Genome Informatics (MGI) database respectively), global regulation of all the genome components (about 228,000 mapped genes in *H. sapiens* and 319,600 in *M*. *musculus*, according to the Ensembl and MGI database respectively) is not achieved by transcription factors (TFs) alone because of their low number (approximately 1500 in mammals [4]) compared to genes to regulate (90% of the genome is transcribed in human [5]). A single TF is estimated to regulate only ten thousand genes according to Chromatin immunoprecipitation and DNA sequencing experiments (ChIp-seq) [6]. Moreover, the involvement of non-coding RNAs in gene expression regulation was evident [7,8]. The evidence that the non-protein coding component of the human genome is actively transcribed and carries out crucial functions limits the importance of coding genes in genome regulation. Non-coding transcripts become one of the stars of modern biology, especially because of their involvement in a wide range of regulatory processes and changed the association of non-coding regions to junk DNA [3].

The genome of multicellular eukaryotes is mostly comprised of non-coding DNA (less than 2% of the human genome codes for proteins [9] while 90% of the human genome is transcribed). Non-coding DNA is transcribed into different classes of non-coding RNAs (ncRNAs) that include structural RNAs (rRNAs and tRNAs) and regulatory RNAs [10]. rRNAs and tRNAs are involved in mRNA translation and can be long or short in size. Regulatory RNAs are further divided into three classes: Small, medium, and long non-coding RNAs. Small non-coding RNAs are between 20 and 50 nucleotides long and include microRNAs (miRNAs) that participate in post-transcriptional regulation [11], small interfering RNAs (siRNAs) that are double-stranded RNAs involved in gene silencing through RNA interfering pathway [12], piwi interacting RNAs (piRNAs) that represent the largest class of small non-coding RNAs in animal cells and regulate genetic elements in germ cell lines such as transposons [13], cis-regulatory RNAs (cisRNAs) [10], centromere repeat-associated short interacting RNAs (crasiRNAs) that are proposed to be essential for cellular stability and chromosome segregation [14], telomere specific small RNAs (telsRNAs) that are heterochromatin associated pi-like small RNAs [15]. Medium non-coding RNAs are between 50 and 200 nucleotides long and include small cytoplasmic RNAs (scRNAs) [16], small nuclear RNAs (snRNAs) [17] that are in splicing speckles and Cajal bodies of the cell nucleus in eukaryotic cells and are involved in pre mRNA processing, small nucleolar RNAs (snoRNAs) [18] that guide rRNA modifications, transcription initiation RNAs (tiRNAs) that modulate local epigenetic structure to regulate transcription factors localization [19], and promoter-upstream transcripts (prompts) transcribed from genomic regions located approximately 0.5 to 2.5 kilobases upstream of active transcription start sites [20]. The remaining non-coding transcripts, longer than 200 nucleotides, are defined as long non-coding RNAs (lncRNAs) (Figure 1).

The number of genes transcribed into ncRNAs varies between species and, interestingly, the complexity of organisms seems to be associated with their abundance implying the potential importance of ncRNAs [1,7]. Life may depend on the transcription of ncDNA because of the role of ncRNAs in genetic regulation. In fact, ncRNAs provide a faster and more flexible regulation in comparison with proteins [7].

However, the precise molecular mechanisms, the biological functions, and the overall role of this huge amount of RNAs in cells remain largely unknown. This is particularly true for lncRNAs that have cell-type specific functions that also depend on the subcellular localization.

Tissues are composed of different kinds of cells and this represents a problem when studying the expression of both genes and proteins by using a standard bulk approach. In fact, the measured expression of a specific gene is the weighted mean of its expression in all the different cells that compose the tissue. The single-cell analysis could overcome this problem by opening new opportunities in the fields of developmental biology [21,22,23], cancer biology [24,25,26], immunology [27,28,29], and neurology [23]. Because of the cell specificity of non-coding RNAs, single-cell approaches may allow a better comprehension of their function and of their involvement in complex processes associated with cell–cell communication. Moreover, single-cell analyses may allow understanding if ncRNAs can determine the generation of cell subtypes as evidenced by single-cell RNA sequencing (scRNA-seq) [30,31,32]. 

In this review, we will focus on lncRNAs because they are key molecules in gene expression regulation. Additionally, most of the reports on ncRNAs in single-cells focus on lncRNAs because the methods used to retrotranscribe RNA target only poly-adenylated RNAs (mRNAs, lncRNAs transcribed by RNA polymerase II) through a complementary oligonucleotide composed by a series of deoxy-thymines (oligod(T)). First, we will describe the lncRNA classification and known functions. Then, we will introduce single-cell analyses and discuss methods available for scRNA-seq and other approaches used for the analysis of lncRNAs at a single-cell level focusing on their pros and cons. We will present results obtained using these methods and discuss what new paradigms have been proposed. To conclude this review, we will present available databases that include information about lncRNAs and variations in the genome, transcriptome, and proteome of single-cells that could be useful for scientists that are planning studies in which lncRNA and single-cell information can be advantageous for the experimental design.

## 2. Classification of LncRNAs according to Genomic Position

The term “long non-coding RNA” (lncRNA) refers to a very heterogeneous group of genes/transcripts. In association with our poor knowledge about the biological functions of several lncRNAs they also do not conserve a primary structure among species, a feature that can be helpful to understand their role. For these reasons, there are different methods of classification, each of them focusing on a different feature of these genes.

The name lncRNA is relatively recent. The first non-coding transcripts studied were classified only based on their role (rRNA, tRNA, cRNA). In the early 2000s, when the abundance of non-coding transcripts became evident [2], new types of classification were introduced.

It is known that lncRNAs represent the largest part of the transcriptome of mammalians [33,34] and plant organisms [35], but only a small fraction of them have been functionally characterized. It has been shown that these transcripts have very heterogeneous roles and mechanisms of action mediated by their cell-specific and subcellular localization. LncRNAs can be classified according to their genomic location or subcellular localization. 

### 2.1. Long Intergenic Non-Coding RNAs (LincRNAs)

Long intergenic non-coding RNAs are transcribed from genomic regions between coding genes; therefore, lincRNAs are not inserted in any protein-coding loci (Figure 2A). LincRNAs are transcribed similarly as mRNAs by RNA pol II and are not usually conserved across multiple species, except those that code for small peptides [36]. LincRNA transcription is associated with chromatin signatures that marks actively transcribed genes such as K4–K36 domains [37]. Several lincRNAs are associated with chromatin-modifying complexes. Rinn and colleagues reported that about 40% of lincRNAs expressed in HeLa cells, human foreskin (hFFs), and human lung (hLFs) fibroblasts were associated with protein complexes involved in chromatin-modifying enzymes (polycomb repressive complex (PRC2) and corepressor for REST (CoREST)) [37] suggesting their involvement in the regulation of gene expression through epigenetic mechanisms. Several transcripts identified as lincRNAs were annotated in the human (17,908 according to Ensembl GRCh38.p13), mouse (100,074 according to Ensembl GRCm38.p6), rat (3090 according to Ensembl Rnor_6.0), zebrafish (1493 according to Ensembl GRCz11), and maize (2532 according to Ensembl Plants B73 RefGen_v4) genomes supporting the idea that they may have important functions in numerous organisms. LincRNAs can also be further divided into groups depending on their transcription orientation compared to the closest protein-coding gene. They can be i) transcribed in the same sense of the nearest gene, ii) transcribed in the opposite direction (divergent), and iii) transcribed in an end-to-end manner (convergent); which is the opposite of divergent (Figure 2A). 

### 2.2. Genic LncRNAs: Intronic and Exonic

LncRNAs can also be localized in a genomic region that overlaps protein-coding genes. These transcripts are called genic lncRNAs. 

Genic lncRNAs could be intronic when entirely overlap with an intron of a protein-coding gene or exonic when entirely or partially overlap with exons of protein-coding genes. In these cases, the transcriptional orientation of genic lncRNAs can be in sense or antisense compared to the transcriptional orientation of protein-coding genes. Genic antisense lncRNAs are transcribed from the opposite strand of the closest protein-coding gene and can overlap with a 5′ head-to-head divergent orientation (SINEUPs [38,39,40,41]) or with 3′ tail-to-tail convergent orientation (3′ overlapping lncRNAs) (Figure 2B). An example of intronic lncRNA is *COLDAIR* which is transcribed from the first intron of the coding gene *Flowering Locus C* (FLC) [42]. It is required for the vernalization-mediated epigenetic repression of FLC itself.

### 2.3. Splicing based Classification

Different RNAs are transcribed by different RNA polymerases (RNA Pol): Transfer RNAs (tRNAs) are transcribed by RNA Pol III, ribosomal RNAs (rRNAs) are mostly transcribed by RNA Pol I and Pol III, while most RNAs are transcribed by RNA Pol II. The latter one synthesizes for messenger RNAs (mRNAs), microRNAs (miRNAs), small interfering RNAs (siRNAs), small nuclear RNAs (snRNAs), small nucleolar RNAs (snoRNAs), piwi-interacting RNAs (piRNAs), and most lncRNAs [43,44]. Some lncRNAs are transcribed by RNA polymerase III [45]. After the transcription step, lncRNAs might be processed by the splicing machinery giving rise to different types of lncRNAs: i) macro lncRNAs that are several kilobases in size and originate from unspliced transcripts, ii) retained intron lncRNAs that are an alternatively spliced transcript of coding genes that lose their coding properties after an intron is retained during the splicing of the transcript (Figure 2C).

## 3. Classification of LncRNAs as Specified by Their Function

### 3.1. Ribosomal RNAs

Historically, first long non-coding transcripts described were rRNAs thanks to their abundance in cells. They are the major structural constituents of the ribosome and can interact with specific sequences of mRNAs (Figure 2D). Prokaryotic ribosomes contain three different RNA molecules while eukaryotic ribosomes contain four. rRNAs are characterized by their sedimentation coefficient (S); prokaryotes rRNA are the 5S, 16S, and 23S while eukaryotes rRNAs are 5S, 5.8S, 18S, and 28S. 5S and 5.8S are small/medium non-coding RNAs because they are 120 and 150 nucleotides long, respectively. On the other hand, 16S, 23S, 18S, and 28S are long non-coding RNAs. 18S is 2100 nucleotides long, 28S~5050 nt, 16S~1.5 Kb, and 23S~2.9 Kb [46,47]. In both prokaryotes and eukaryotes, rRNA genes are transcribed as a single large pre-rRNA molecule (16S, 23S, 5S rRNA in prokaryotes and 18S, 28S, and 5.8S in eukaryotes) and then processed to produce the single rRNAs. In eukaryotes, 5S RNA is transcribed by RNA polymerase III [45] while 5.8S, 18S, and 28S RNAs are transcribed by RNA polymerase I [48].

### 3.2. Chromatin Interacting RNAs

In the late 1960s, James Bonner introduced and described a distinct class of RNAs capable of binding chromatin: chromosomal RNA or cRNA [49]. LncRNAs can interact with chromatin in multiple ways; the most common being the recruitment of the polycomb repressive complex (PRC). PRC induces chromatin modifications and consequently epigenetic based silencing of genes. Polycomb proteins form two major PRC: PRC1 and PRC2. PRC1 components were first characterized in Drosophila [50] and then, homologs genes were identified in human: CBXs (polycomb homolog), PHC1, 2, and 3 (polyhomeotic homologs), Ring1a and Ring1b (dRING homologs) BMI1 (Polycomb Ring Finger Proto-Oncogene) and six minor others (posterior sex combs homologs) [51]. Functionally, PRC2 binds to chromatin according to DNA CpG density and methylation status. PRC1 may indirectly participate in the localization of PRC2 in unmethylated CXXC DNA domains guiding H3K27me3-mediated chromatin silencing [52]. PRC2 can bind to unmethylated DNA independently of PRC1 via PRC2-accessory proteins with DNA binding capacity, such as transcription factors both in Drosophila and mammalian [53,54,55,56,57]. Other methods are involved in polycomb group (PCG) positioning along chromatin (see the review [58]), but the most important for this review is that based on RNA. Several studies have reported the binding of PRC2 to lncRNAs such as XIST and repA in the inactivation of the X chromosome in mammals [59] and HOX Transcript Antisense RNA (HOTAIR) lncRNA in silencing of hox genes in human [60,61] (Figure 2E).

### 3.3. miRNA Sponges

LncRNAs can interact with miRNAs to act as post-transcriptional regulators of protein expression. miRNAs are short non-coding RNAs (18–28 nt) [62] that interact with target mRNAs for their cleavage or to simply repress their translation [63,64]. LncRNAs that interact with miRNAs are classified as competing endogenous RNAs (ceRNAs). This definition was proposed by Salmena and colleagues in 2011 [65]. ceRNAs are also known as miRNA sponges and compete with mRNAs for miRNA binding (Figure 2F). One example of ceRNA is the long intergenic non-coding RNA-Muscle Differentiation 1 (linc-MD1). This is a muscle-specific transcript of about 500 nucleotides that contains numerous target sites for miR-133 and -135 that regulate Mastermind Like Transcriptional Coactivator 1 (MAML1) and Myocyte Enhancer Factor 2C (MEF2C), transcription factors that activate muscle-specific gene expression [66]. Another important class of lncRNAs acting as miRNA sponges are pseudogenes. An example of pseudogene showing miRNA sponge activity is Phosphatase And Tensin Homolog Pseudogene 1 (*PTENP1*). The protein-coding gene Phosphatase And Tensin Homolog (*PTEN)* is regulated by miR-19b and -20a which can be sequestered by the pseudogene PTENP1 affecting the level of its cognate gene [67]. Another example of miRNA sponging RNAs are circular RNAs (circRNAs). They form a covalently continuous loop characterized by the end terminal joining having on their sequence multiple binding sites for a specific miRNA. For a specific review on circRNAs see [68]. 

### 3.4. Enhancer RNAs 

LncRNAs transcribed from enhancers are called enhancer RNAs (eRNAs). Enhancers are short DNA regions (50–1500 bp), usually isolated from coding genes, that can influence gene transcription by binding specific protein activators [69]. eRNAs can increase the expression of target genes in cis and were first described in 2010 using RNA-seq and Chip-seq techniques [70]. Enhancer elements were first described by Banerji and colleagues in 1981 noticing how the expression of a beta-globin gene is enhanced by remote SV40 DNA sequences [71]. Other known eRNAs are Alpha-250/Alpha-280, Evf-2 and ncRNA-a1 [72]. The mechanism of transcriptional enhancement can vary, some eRNAs increase the strength of the enhancer-promoter looping while others will impede the binding of negative elongation factors (NELFs) to the promoter to reduce transcriptional repression [73,74]. eRNAs can be classified as 1D (unidirectional) or 2D (bidirectional) depending on their transcription direction. In the first case, eRNAs are longer (>4 kb) and polyadenylated, while in the second case they are shorter (0.5–2 kb) and non-polyadenylated [75,76] (Figure 2G).

### 3.5. SINEUPs 

SINEUPs are modular antisense lncRNAs that have two sequence specificities: they contain an inverted SINEB2 sequence and a small complementarity sequence with the targeted mRNA. Due to these characteristics, SINEUPs can up-regulate the translation of mRNAs in a gene-specific manner without affecting gene expression. SINEUPs are coded by genes that overlap with the target protein-coding gene to allow a 5′ head-to-head binding [77,78,79] (Figure 2H). SINEUPs can also be designed artificially and can be used to obtain the opposite effect of miRNA/siRNA translational silencing.

### 3.6. LncRNAs Coding for Micropeptides

Analyzing the ability of ribosomes to associate with lncRNA molecules [80] researcher concluded that some lncRNAs can be translated [36,81] (Figure 2I). Although ribosome profiling supports the interaction of lncRNA with ribosomes the effective presence of small peptides is not guaranteed. It is important to associate a mass spectrometry analysis (MS) to confirm the presence of translated peptides. For example, among 233 lncRNAs associated with ribosomes in human cell lines, only 18 were confirmed to produce small peptides by MS [82], in Zebrafish results were similar. Among 535 lncRNAs with coding properties, 6 were confirmed with MS [83]. Interestingly, many micropeptides were functionally characterized evidencing their importance in muscle physiology regulating both muscle regeneration via mTOR [84] and development [85], in the regulation of cancer metabolism [86], in signal transduction [87]. See Appendix A for a list of functionally characterized micropeptides originating from lncRNAs. For a more completed review on micropeptides from lncRNAs and methods used to study them see [88]. 

### 3.7. Target Position

Some chromatin interacting lncRNAs act by targeting the chromatin region from which they are transcribed. These are called cis-acting lncRNAs. The targeted chromatin is not necessarily restricted to the close proximity of the lncRNA, for example, the well-known lncRNA XIST acts by targeting the whole X chromosome in cis (Figure 2J).

On the other hand, if the lncRNA target is not located in the proximity of its gene it is described as a trans-acting lncRNA (Figure 2J). HOTAIR is a trans-acting lncRNA that in humans is transcribed in chromosome 12 and then transported by the Suz-Twelve protein to its target site, the homeobox D cluster on chromosome 2. Trans-acting lncRNAs can bind to transcription elongation factors or other proteins like the RNA polymerase and can have multiple targets throughout the genome. For a review on gene expression regulation by cis-acting lncRNAs see [89].

For a summary of functionally defined lncRNAs see Appendix A where web references are linked to easily retrieve the available information about the lncRNA of interest.

## 4. Classification of LncRNAs according to Their Subcellular Localization 

The role of lncRNAs is strictly dependent on their subcellular localization. For example, chromatin interacting lncRNAs need to be localized in the nucleus while miRNA sponges are most commonly found in the cytoplasm.

### 4.1. Nuclear LncRNAs

LncRNAs that are only found in the nucleus are classified as nuclear lncRNAs. These lncRNAs contain specific sequences to indicate that they have to be retained in the nucleus [90,91]. Most of the nuclear lncRNAs are regulators of transcription or influence mRNA processing; they can both enhance or silence the transcription of genes, for example, by recruiting transcription factors or by acting as decoy impeding the binding of transcription factors to DNA. One example is the lncRNA Breast Cancer Anti-Estrogen Resistance 4 (BCAR4) that influences cell migration in breast cancer by binding to Smad Nuclear Interacting Protein 1 (SNIP1) and Serine/threonine-protein phosphatase 1 regulatory subunit 10 (PNUTS), two transcription factors that induce the activation of a non-canonical hedgehog/GLI2 (GLI Family Zinc Finger 2) transcriptional program [92]. Nuclear lncRNAs include the most studied lncRNAs such as XIST [93], MALAT1 [94], and NEAT1 [95].

### 4.2. Cytoplasmic LncRNAs

LncRNAs that localize into the cytoplasm are classified as cytoplasmic lncRNAs. The transport of lncRNAs such as mRNAs from the nucleus to the cytoplasm involves several different proteins localized on both sides of nuclear pore. The complex composed of the Tho complex, the RNA helicase UAP56 (or its paralog URH49) and Aly is loaded onto the mRNA or lncRNA in the nucleus producing the RNA protein complex (RNP). Then, the nuclear export receptor composed of Nuclear RNA Export Factor 1 (Nxf1) and Nuclear Transport Factor 2 Like Export Factor 1 (Nxt1) (also known as TAP and p15) associates with Aly and allows the transport at the nuclear pore of its cargo physically interacting with the phenylalanine (F) and glycine (G) repeats of proteins in the interior of nuclear pore. After the transport through the nuclear pore the RNP is remodeled to remove certain nuclear associated exported factors, recycled back into the nucleus, and to make the mRNA or the lncRNA stably located in the cytoplasm [96].

Cytoplasmic lncRNAs are more commonly involved in post-transcriptional regulation and as previously described, competing endogenous RNAs can act as alternative targets for miRNAs to reduce the amount of RNA interference while SINEUPs, another family of cytoplasmic lncRNAs, can enhance the translation of target mRNAs without affecting mRNA transcription. Another example of cytoplasmic lncRNA is Tissue Differentiation-Inducing Non-Protein Coding RNA (TINCR) that binds to mRNA through a 25 nt sequence. This lncRNA has a strong affinity for the protein STAU1, the complex formed by this interaction is known to mediate the stabilization of mRNAs involved in epidermal differentiation [97]. 

### 4.3. Mitochondrial and Chloroplastic LncRNAs

Eukaryotic cells have specific organelles for energy production (mitochondria) or photosynthesis (chloroplasts) in the case of cells of plants. Both organelles have endosymbiotic origins and present their own genome that during the evolution has lost most of the genes leaving to the nuclear genome the task of synthesizing for most of the proteins for organelles functions. Mitochondrial genome maintains the capacity of synthesizing for 13 proteins (required for part of OxPhos complexes), 22 tRNAs, and two ribosomal RNAs (12S and 16S). Mitochondrial DNA is a double-stranded circular molecule where it can be distinguished a heavy strand (rich in guanine) and a light strand. Heavy strand is reach in genes while light codes only for one subunit of nicotinamide adenine dinucleotide dehydrogenase (ND), 8 tRNAs, and 3 lncRNAs. Interestingly, these lncRNAs regulate ND5 (NADH dehydrogenase subunit 5), ND6 (NADH dehydrogenase subunit 6), and CYTB (cytochrome b) throughout complementary binding of respective RNAs [98]. Nuclear-encoded mitochondrial proteins regulate their expression confirming the intercommunication among nucleus and mitochondria for their function. Previously described lncRNAs function in mitochondria, but it also synthesizes for lncRNAs that can be transported in the cell nucleus. For example, a chimeric RNA composed by 16S mitochondrial RNA plus an inverted repeat of 120 bp was localized in nuclei of mouse and human sperm [99,100] once more confirming the importance of intercommunication among these “organelles”. Extramitochondrial localization of the 16S mitochondrial rRNA was demonstrated in Xenopus and Drosophila [101,102] where it was demonstrated involved in the generation of progenitors of the germline [103]. Mitochondrial DNA allows the synthesis of two lncRNAs (sense noncoding mitochondrial RNA or SncmtRNA and ASncmtRNAs) related to cell proliferation [104] and tumor suppression [105] and LIPCAR (long intergenic noncoding RNA predicting cardiac remodeling) associated with the risk of heart failure [106]. Other lncRNAs were synthesized by the nuclear genome and transported in mitochondria such as Survival Associated Mitochondrial Melanoma Specific Oncogenic Non-Coding RNA (SAMMSON) [107]. 

The genome of chloroplasts, such as mitochondrial one, synthesizes for lncRNAs. Hotto and colleagues identified 107 non-coding RNAs from *A. thaliana* chloroplasts by RNAseq proposing a regulatory action by matching to the 5′-UTR, 3′-UTR, or coding regions of specific targets [108]. See [109] for a review on the function of non-coding RNA in plant organelle biogenesis.

## 5. Methods for Transcriptional Analysis of Single Cells: Progresses and Limitations

Since the first reports of single-cell RNA-sequencing in 2009 [110,111], novel strategies have been developed by the scientific community to increase the efficiency of the methods and achieve high-resolution results. The main steps for the transcriptional analysis of single cells include (1) single-cell isolation, (2) sequencing library preparation and (3) RNA sequencing and bioinformatic analyses.

### 5.1. Single-Cell Isolation 

To obtain a cell suspension, the enzymatic digestion or mechanical disruption of tissues are commonly used as isolation methods. Depending on the biological question to be addressed and on the type of starting material, a pool of cells can be enriched based on its particular morphophysiological features (e.g., size, shape, electrical charge, expression of molecular markers). Different technologies can be used for enrichment, such as microfiltration, density gradient centrifugation, immunoaffinity, and magnetic-activated cell sorting [112]. However, the major issue of pre-analytical processing is the disturbance of native expression profiles of cells which can lead to misinterpretation of data. Thus, reducing the processing time with minimal manipulation of samples is still a challenge to overcome. Several methods have been established to isolate single cells, each with particular benefits and drawbacks. The micropipette isolation method consists in diluting the cells to a very low concentration and seeding them into a tube [113,114] or 96 well plate [115] (Figure 3A). This technique does not require sophisticated platforms but is laborious and not very efficient. The laser capture microdissection (LCM) consists of coupling a laser system (infrared or ultraviolet) to a microscope that can capture a single cell from a solid tissue [116,117] (Figure 3B). The advantages of LCM include visual inspection and spatial information of cells on the tissue and the possibility to store the remaining tissue. Even though this technique does not require enzymatic digestion, tissues must be stabilized using fixatives, which can reduce RNA quality and lead to biased results. Moreover, during LCM, heat induced by the laser may degrade RNA. 

The fluorescent activated cell sorting (FACS) has higher efficiency than the previous methods and the principle of this technique is based on tagging cells with a fluorescent monoclonal antibody specific to surface antigens or intracellular molecules. This allows the instrument to separate positive cells from negative cells and debris [118] (Figure 3C). Besides, negative selection can be carried out on unmarked cells. This method allows the analysis of specific cells of interest, which can reduce the overall experiment cost. The limitation of this technique is the large input (more than 10,000 cells) and the necessity of antibodies to target desired proteins. The capillary-based single-cell collection, employed by AVISO CellCelector (ALS) [119], integrates robotic technology and image processing software that allows the capturing of individual cells by mechanical suction and seed them on individual wells of a culture dish [120] (Figure 3D). Images can be acquired before and after single-cell isolation to monitor efficiency, but this system is not recommended to work with cells that have strong adhesive properties. The punching technology consists of filling samples into chips or arrays, collect individual cells into microwells, select cells of interest (based on fluorescence signaling) and then “punch” them into PCR tubes or plates for further analysis (Figure 3E). There are two main platforms available: Puncher Platform (Vycap, Eindhoven, Netherlands) and CellRaft AIR System (CellMicrosystems, Cary, NC, USA). The first one can process up to 50 mL samples and is optimized for the isolation of rare cells in large volumes, such as circulating tumor cells (CTC) and fetal trophoblasts [121]. The second one is designed to collect both the magnetic microwell and the single cell attached to it by using a magnetic wand. This mechanism has minimal disturbance on cells during the capture and is optimized for adherent cells [122]. The dielectrophoresis (DEP) technique is performed using the DEPArray NxT instrument (Menarini Silicon Biosystems, Bologna, Italy). This technology explores the electrokinetic properties of suspended cells to create “electric cages” and capture them with a non-uniform electric field. When a DEP cage is moved by a change in the electric field pattern, the trapped cell moves with it. The system allows visual control of selected cells and the low voltage applied does not interfere with cell viability [123] (Figure 3F). Microfluidic chip platforms can encapsulate single cells with a bead inside aqueous droplets in an oil phase [124]. Each bead contains thousands of DNA copies with a poly(dT) expansion to capture polyadenylated RNAs and unique DNA sequences (barcodes) that can be used to identify the correspondent cell of origin. Moreover, to overcome library construction errors, such as duplications of reads due to different efficiency of cDNA synthesis, the bead can also contain unique molecular identifiers (UMI) sequences, specific to each molecule [125] (Figure 3G). Several platforms use this process to profile gene expression of single cells. Some platforms (e.g., Nadia—Dolomite Bio, Royston, UK), do not contain reagents for reverse transcription inside the droplets and the library preparation is performed after pooling the collected cells and breaking off the droplets. Other instruments (e.g., Chromium System—10× Genomics, Pleasanton, CA, USA; InDrop System—1CellBio, Watertown, MA, USA; C1 System—Fluidigm, San Francisco, CA, USA; ddSEQ Single-Cell Isolator—Bio-Rad-Illumina, Hercules, CA, USA) can perform the reactions inside the droplet, increasing the efficiency of the method. These platforms enable the analysis of up to 10,000 single cells (except the C1 System that process up to 800 cells) in a parallel manner, avoiding cross-contamination and increasing reproducibility [112]. However, the RNA capturing efficiency of these techniques can reach 65% maximum [112] and they require high input and clean samples because the microfluidic channels can clog if there are cell clumps or debris. The current trend is to use instruments that provide end-to-end solutions, such as the C1 System and the Chromium System, that process from the single-cell isolation to data analysis in a single platform. Usually, RNA sequencing is performed with Illumina platforms. In many cases, single cells cannot be used (e.g., when starting from frozen tissues) and therefore single nuclei can be an option. In fact, it was evidenced a good correlation among nuclei gene expression and cell gene expression [126,127]. See Table 1 for a comparison of different systems used to collect single cells. 

### 5.2. Library Preparation

After the isolation step, single-cells have to be lysed and the RNAs must be reverse transcribed maintaining the ability to identify from which cell derives the RNA information. As previously discussed, the number of cells that can be analyzed with single-cell approaches is comprised between 100 and 10,000 cells (Table 1). It is estimated that only 10–20% of transcripts are converted into cDNA with the available protocols. Thus, cell lysis is a critical step to obtain high-quality results [128]. Oligo-dT primers can be used to initiate the cDNA reaction, but they will only anneal on polyadenylated RNAs, such as mRNAs and some lncRNAs. Thus, the analysis of other classes of non-coding RNAs requires specific protocols (e.g., for circularRNAs [129], for total RNA [130,131], or miRNAs [113,132]). There are two main strategies to run the cDNA amplification: The template switching and the in vitro transcription. In the template switching (SMARTer) method, a modified oligo(dT) primer is used to prime first-strand synthesis. Then, the Moloney Murine Leukemia Virus (MMLV) derived reverse transcriptase adds a short deoxycytidine tail at the 3′ end of the first-strand cDNA. Deoxycytidine tail is used for the template-switch primer hybridization through its deoxyguanosine 5′ end. Its hybridization provides a template for the retro transcription (RT) enzyme that extends to the end of the template-switch primer the cDNA, producing single-stranded cDNA having two known anchors at 3′ and 5′ ends. Anchors allow the optimized amplification of polyadenylated RNAs through PCR to produce enough DNA to perform a successful sequencing even starting from just a few pg of total RNA [133,134]. This straightforward approach allows uniform coverage. 

On the other hand, the linear amplification is based on the in vitro transcription (IVT) by the T7 RNA polymerase, that synthesizes multiple copies of RNA, followed by another round of reverse transcription [135]. This additional step is time-consuming and can introduce 3′ coverage biases. Moreover, all amplification steps were largely discussed because of their ability to maintain a linear amplification range to preserve gene expression proportions [136,137].

### 5.3. RNA Sequencing and Bioinformatic Analysis

Final steps of single-cell RNA (scRNA) analysis are the sequencing and bioinformatic analyses. Alternatives to RNA sequencing are qRT-PCR and microarrays. The former is laborious and has a limited number of targets, while microarrays demand previously established transcriptomic probes and might have some background noise interfering on low abundant transcripts [138,139]. Thus, the available gold standard method is scRNA-seq. Once sequencing reads are obtained, low-quality bases, adaptors, and barcodes need to be trimmed before alignment. Spike-in controls can be used as quality control and also to estimate the absolute levels of RNA expression since in scRNA-seq the exact number of cells is known [140]. After alignment, reads are annotated in exonic, intronic, or intergenic features. Then, normalization is performed to avoid cell-specific bias. In fact, a problem of scRNA-seq is the presence of zero-inflated counts in the data expression matrix. For a review on scRNA-seq data processing see [141]. To increase the sequencing coverage of low input samples in a cost-effective manner, alternative methods focusing on the sequencing of 5′ or 3′ ends of transcripts were developed. However, it is not suited for allele-specific expression or isoform usage [142]. Numerous tools have been developed to analyze scRNA-seq data, due to its intrinsic features. There are at least 493 tools available on the web [143], covering over 30 categories including quality control, read alignment, normalization, and gene networks analysis. Concluding, scRNA-seq can be performed on full-length RNA or just on its 5′ or 3′ ends with the consequent necessity of performing the sequencing at a different depth. Currently full-length transcript libraries are sequenced at the depth of 10^6^ reads per cell with the possibility to increase the sequencing depth (15–25 × 10^6^ reads per cell) if interested in RNA alternative splicing. Differently, by sequencing 3′ end libraries researchers can use a lower depth sequencing (10^4^–10^5^ reads per cell) [144]. It is interesting to highlight the fact that generally the average number of lncRNAs identified by scRNA-seq is lower than that of mRNAs and this could be due to their lower expression in respect to coding genes [145,146]. 

## 6. Single-Cell Analysis of LncRNAs

As lncRNAs keep getting increased attention, it has become clear how their localization, tissue, and cell specificity are key factors in their role as regulatory elements. Many studies have described the developmental and tissue-specific expression of these transcripts in different organisms (eg. human, mouse, or fruit fly) [147,148,149]. However, it was only recently that it became possible to study lncRNAs at the single-cell level with a genome-wide approach by using scRNA-seq.

According to our search, the first work measuring the expression of a lncRNA at a single-cell level was published in 2003 by Hartshorn and colleagues [150]. They took advantage of the possibility of isolating a single blastomere manually from 8 cells-stage embryos, collect it in a tube, dissolve the cell, and analyze DNA or RNA by PCR. The expression of the lncRNA XIST and the coding RNA Sry were detected thanks to specific molecular beacons. Interestingly, they showed that the expression of XIST varied in different cells supporting the idea that cells in an early mammal embryo are not all functionally equivalent. In 2005 the same group ameliorated the method to use SYBR Green based detection method in the PCR reaction instead of molecular beacons [151]. In both cited studies Hartshorn and colleagues co-amplified and simultaneously quantified RNA and DNA copies of two specific genes (*XIST* and *Sry*) having sexually distinct patterns of expression in the early embryo. Female samples, that have two X chromosomes, always result in XIST amplification but no Sry amplification, while male embryos, that have one X and one Y chromosome, always generated equal numbers of XIST and Sry amplicons since the *Sry* gene is on the Y chromosome. In 2007 they developed a new method to quantify different genes whose expression is not functionally guided by sex. Using the Linear-After-The-Exponential-PCR technique (LATE-PCR [152]) they demonstrated both XIST and Oct4 were expressed to variable degrees in the cells of the same embryo [153] (Figure 4A). In this work, Hartshorn and colleagues demonstrated, by analyzing single blastomeres, that Oct4 is expressed in all cells of a human 8-cell embryo, not from a specific cell sub-population, as demonstrated by gel PCR based analysis [154]. Works developed by Hartshorn demonstrated that both coding and non-coding genes have an expression that is cell-to-cell dependent and support the necessity of having a picture as detailed as possible of molecules expressed by each cell type to understand its interactive network in a tissue.

### 6.1. LncRNAs in Embryo-Derived Cells

XIST is one of the best known lncRNAs [93]. To better understand the process of X chromosome inactivation, the expression of XIST was analyzed in single cells of human embryos through the oocyte-to-embryo transition and in new mRNA reprogrammed iPSCs by [157]. Briggs and colleagues used a combination of FISH, qRT-PCR, and FACS to show that XIST is initially expressed in newly reprogrammed iPSCs. They also highlighted that the expression of this lncRNA is lost after some passage of cells and that this affects the expression of X‑chromosome genes. This and previously presented works were based on specific lncRNAs. The development of scRNA-seq techniques allowed for more broad studies firstly of coding and then of non-coding RNAs. Thank this technical development it was possible to evidence that several genes, including lncRNAs, can escape from the X chromosome inactivation [158] (Figure 4B).

In 2013 Yan and collaborators systematically analyzed for the first time maternally expressed lncRNAs in human early embryos single cells. They used the micromanipulation to isolate single cells. In their work, they generated sequencing data from 124 single cells using an Illumina HiSeq2000 sequencer. In particular, they found 7214 known lncRNAs and 1487 novel lncRNAs maternally expressed in human mature oocytes [159]. scRNA-seq quickly became one of the most common techniques for single-cell studies in both human and mouse especially to study embryonic stem cells because of their biological importance and the possibility to physically isolate them.

Petropoulos and colleagues generated a transcriptional database analyzing 1529 individual cells from 88 human preimplantation embryos [155]. They used a large number of embryos in order to dilute out embryo-specific differences that might be caused by embryo-specific factors. Their analysis showed how transcriptomes are primarily segregated based on the embryonic stage and then on the lineage, embryo-to-embryo variability, and subpopulation. In humans, the X chromosome inactivation may be compatible with a partial and complementary inactivation of both X chromosomes in females. In fact, the authors evidenced that pluripotent ground-state should be characterized by female cells expressing XIST also having both X chromosomes active and female to male dosage compensation (Figure 4B).

Cellular fate and differentiation during development have been investigated by different researchers with a single-cell approach [160]. In this context, Rizvi and colleagues proposed an algorithm to study transcriptional regulation to identify cell identity over time and therefore cell differentiation in response to specific stimuli. The proposed algorithm, called single-cell topological data analysis (scTDA), does not work as previously developed algorithms that suppose a one or three-like branched structure, but it is based on an unsupervised method that has the advantage of extracting more complex relationships [143]. An interesting feature of this algorithm is that it was able to identify cell states by using lncRNAs supporting the notion of their cell stage-specific expression. The authors showed that applying this algorithm to analyze murine embryonic stem cells they were able to identify four transient states (pluripotent, precursor, progenitor, and fully differentiated cells).

Recently, two different techniques were developed for the analysis of total RNA from single cells [130,131]. Compared to other methods that use oligo(dT) in association with SMART (acronym for A Simple Method for Amplifying RNA Targets) amplification or the activity of deoxynucleotidyl transferase to create an anchor in the 3′ end of cDNA [161], random displacement amplification sequencing (RamDA-seq) and multiple annealing and dC-tailing-based quantitative single-cell RNA-seq (MATQ-seq) show higher sensitivity to non-polyadenylated RNA and near-complete full-length transcript coverage without no significant end bias. Using RamDA-seq, Hayashi and colleagues were able to reveal 7580 transcripts differentially expressed from embryonal stem cells, including 458 non-polyadenylated transcripts some of which were lncRNAs (e.g., Neat1). Before the RamDA-seq development, Tang and colleagues analyzed the expression of mouse blastomere single cells obtaining full-length RNA sequences [162,163] that were re-analyzed by Zhang and colleagues to identify 5563 novel lncRNA transcripts from 3492 loci expressed in mouse cleavage stage embryos [164]. Interestingly, the identified lncRNAs tend to be expressed in a specific developmental stage manner [165] probably because they are involved in the regulation of peculiar mechanisms typical of specific developmental stages. 

### 6.2. LncRNAs in Stem Cells

LncRNAs have been studied at the single-cell level not only using Embryonic Stem Cells but also using other types of stem cells or pluripotent cells.

Johnson and collaborators used a combination of FACS, RNA-sequencing, and PCR on single cells to study neuronal progenitors at the single-cell level [166]. They showed how human radial glial cells (RGC) differ from mouse RGC by having a more “gradual” transition to neuronal lineage-committed intermediate progenitors. They also identified several previously no described lncRNAs enriched in human outer radial glia (ORG) and found that many of these genes show highly distinct expression patterns in ORG cells. This suggests that lncRNAs participate to the definition of molecular identity and function of the ORG subpopulation in humans. The authors identified 75 putative lncRNAs in human samples, but only 18 appeared conserved among human and mouse supporting data that describe lncRNAs as non-conserved transcripts. Also the lncRNA LOC646329 is involved in neuronal cell proliferation. These data support the involvement of lncRNAs in neuronal developing processes and sustain the necessity to face also on them in the future when researchers discover molecular mechanisms involved in the development and modulation of neurological pathologies. 

The long non-coding RNA H19, in association with XIST, is one of the most studied lncRNA. In the field of single stem cell analyses, it was studied in embryonic hematopoietic stem cells (HSC). By combining RNA single-cell profiling and functional screenings Zhou and colleagues [145] showed that pre-HSCs not expressing the lncRNA H19 failed to generate HSCs. The authors showed that the action of H19 was not dependent on the miR-675, coded from the same locus, but by the increased activity of S-adenosylhomocysteine hydrolase that is a regulator of DNA methylation. As a consequence of H19 silencing, pre-HSCs showed increased DNA methylation on promoters associated with the downregulation of several transcription factors involved in hematopoiesis such as Runx1 and Spi1.

In 2015 Kim and colleagues published a research where, using single-cell RNA sequencing with the SMART-seq method (Illumina), they described the dynamic expression of 437 lncRNAs during the reprogramming of fibroblasts to pluripotent cells. The authors showed that the activation of many lncRNA is an early molecular event in early stage iPSCs. Moreover, they showed that some lncRNAs are functionally relevant for reprogramming [156] (Figure 4C). In addition to lncRNAs, pseudogenes also appeared to be important in the reprogramming process, supporting the necessity of further analyses on non-coding RNAs in different reprogramming contexts to advance our basic knowledge of cell state plasticity and cellular transformation. 

### 6.3. LncRNAs in Differentiated Cells

Rather than using pluripotent cells, several studies have instead focused on specialized/differentiated cells. In 2015 Cabili and colleagues developed single-molecule RNA-FISH to quantify the subcellular localization of lncRNAs in three different cell types: hFFs, hLFs, and HeLa cells [167]. They were able to show a variety of different localization patterns, including bright sub-nuclear foci that dissolve during mitosis. They based the lncRNA detection on the hybridization of different fluorescent-labeled probes constructed along the lncRNA sequence. In fact, lncRNAs have specific 3D structures that cause problems in the prediction of working probes. The use of different probes spanned along the sequence of the same transcript may overcome this problem. This work was particularly important for developing a medium-throughput technology to correctly evaluate the subcellular localization of lncRNAs. In fact, as previously discussed, it is a fundamental parameter to be considered to understand the function of this complex class of RNAs.

Even if the manually recovering of a single cell is laborious, it can be a good alternative to analyze cells with particular features and identify functionally lncRNAs. In 2017, RNAseq on manually selected neuronal cells was used by Spaethling and colleagues to analyze the expression of single cells purified from the human brain. The authors evidenced that among 935 identified lncRNAs, 113 were differentially expressed throughout the cell types and associated the expression of lncRNAs with marker-based cell types. For example, the long intergenic non-coding 01314 (*LINC01314*) was specifically expressed in astrocytes, the long intergenic non-coding 00152 (*LINC00152*) in endothelial cells, the long intergenic non-coding 00520 (*LINC00520*) in microglia, the Maternally Expressed 3 (*MEG3*) in neurons, the long intergenic non-coding 01105 (*LINC01105*) in oligodendrocytes, the Nuclear Paraspeckle Assembly Transcript 1 (*NEAT1*) was highly expressed in astrocytes and endothelial cells while the Small Nucleolar RNA Host Gene 6 (*SNHG6*) was expressed in all cells analyzed [168].

After manually recovering single muscle fiber, we used microarray technology to compare the expression of lncRNAs in single fast and slow myofibers and to dissect subcellular localization of lncRNAs. We revealed that cells with different metabolic properties express different lncRNAs. Moreover, we revealed that some differentially expressed lncRNAs localize differently in cellular compartments according to cell metabolism and lncRNA isoform [114]. This last result suggests that the same lncRNA could have different roles in different cells and that the same might be true for different isoforms of the same lncRNA. We validated our results by using both qRT-PCR and RNA-FISH, on both C2C12 myoblasts and myofibers. We functionally discussed the involvement of the lncRNA Pvt1 in the modulation of muscle atrophy and mitochondrial network. Our findings suggested that Pvt1 can modulate the expression of c-Myc, which in turn regulates Bcl-2, having an effect on autophagy and apoptosis that are strictly linked to the conformation of the mitochondrial network [114].

### 6.4. LncRNAs in Tumors

Since lncRNAs are involved in cell differentiation it is interesting to study their function in tumors. In fact, it was proposed an intimate link between altered differentiation programs and cancer [169]. Moreover, a tumor may be composed of different cell clones that mutate, compete and evolve resulting in the outcome of a malignant neoplasm [170]. For this reason, it is more informative an analysis of single cells composing cancer instead of the analysis on bulk tissue that usually contains also normal not neoplastic cells.

Wang and collaborators explored the function of lincRNAs in cervical carcinoma formation using Hela-S3 cells as a biological model [171]. They analyzed gene expression from 37 single Hela-S3 cells detecting an average of 511 lincRNAs in each cell. By comparing the expression patterns of the lincRNAs with one of protein-coding genes the authors of this work found that lincRNA expression displays more cell specificity than that of protein-coding genes. This cell expression specificity was also evidenced by Wu and colleagues that analyzed cells from Myelodysplastic syndromes [146]. These are a group of cancers in which patients present immature blood cells that do not complete maturation. The authors inferred the function of lncRNAs by correlating their expression with that of coding RNAs. They demonstrated that lncRNAs were involved in the regulation of hematopoiesis and related cellular functions. Finally, they associated an aberrant expression of lncRNAs with myelodysplastic syndromes supporting their functional role in these pathologies. On the other hand, Pang and colleagues focused on glioblastoma (GBM). They used scRNA-seq based on SMART-seq to analyze a novel parameter they describe as “pseudotime”, which refers to transcriptional changes in cells during GBM progression. They were able to describe the path along which GBM stem cells (GSCs) gradually transform into invasive cells, called the “stem-to-invasion path” [172]. The path contains transcription factors and lncRNAs supporting one more time their importance in cancer development. Analyzing lncRNAs expressed in single cells derived from GBM, Hu and colleagues recovered a very small number of lncRNAs that may participate in the regulation of fundamental cellular processes (2% of lncRNAs in GENECODE) [173]. These lncRNAs may be a target of further studies to evaluate how lncRNAs regulate processes fundamental for cell replication and survival.

Summarizing, scRNA-seq allowed to better characterize processes already known such as X chromosome inactivation, or cell differentiation that also impact on tumor progression. Notably, it is also useful to define new functions for lncRNAs. In fact, the co-expression pattern search was used to identify transcription factors involved in gene expression regulation [174], to recognize cis-regulatory elements [175] and can be useful to infer regulatory functionality of lncRNAs. The identification of correlation between the expression of lncRNAs, mRNAs, or proteins, possibly associated with subcellular localization, gives the opportunity of inferring regulatory functionality of lncRNAs.

## 7. Databases

### 7.1. Collection of Single-Cell Gene Expression

After the development of different techniques to physically separate single cells, single-cell genomic and/or transcriptomic profiles have been accumulated. This huge amount of data is available, usually as raw data, in the Sequence Read Archive (SRA) [176] or in the Gene Expression Omnibus (GEO) [177] databases. Unfortunately, data on these repositories of genome-wide microarray and sequencing experiments are not easy to use and need bioinformatics skills. To make more accessible such complex information, different databases were developed with user-friendly interfaces. The European Bioinformatics Institute (EMBL-EBI) developed a single-cell expression atlas [178] containing 52 experiments for cells from *H. sapiens*, 51 for *M. musculus*, 7 for *D. rerio*, 1 for *R. norvegicus*, 1 for *S. mansoni*, 1 for *A. gambiae*, 1 for *D. melanogaster*, and 1 for *C. elegans*. The EMBL database does not only contain data from animals but also from plants (5 for *A. thaliana*), fungi (1 for *S. cerevisiae*) and protists (2 for *P. berghei*). The Institute of Physical and Chemical Research in Japan (RIKEN) also developed a single-cell centric database [179] to enable easy access of single-cell datasets [180]. The RIKEN database contains experiments performed on *H. sapiens* and *M. musculus* single cells. Broad institute developed its own database for single-cell datasets [181] containing 117 studies and data for about 3,780,000 cells. Several other databases were developed and published. For a comprehensive list of databases presenting single-cell experiment results not described in the text see Table 2. All databases give the possibility to view cell clusters and to search for information starting from a specific experiment. Not all databases have the possibility to interrogate them with a specific gene to recover its expression in all experiments included in the database. This last restriction may be a limitation for the database usage. However, studying a gene or a group of genes in specific tissues trying to understand which cells in that tissue contribute to the activity of studied genes without performing single-cell experiments may be important to formulate new experimental hypothesis.

### 7.2. Databases of LncRNAs

As previously described, lncRNA classification is complex and defined according to their genomic position, function, and subcellular localization. Several databases integrated the knowledge about lncRNAs in different species. All these databases may be functional for researchers that approach this newly discussed RNA molecules. The most known databases for lncRNAs are Ensembl [191], Refseq [192], and GENCODE [193]. Other databases were developed such as Noncode [194] that includes curated information retrieved from literature associated with Ensembl, RefSeq, lncRNAdb, and GENCODE for 17 species, and Lincipedia [195] that includes information only for human lncRNAs. Recently was developed the experimentally validated lncRNA database (EVlncRNAs) [196] containing 1543 lncRNAs from 77 species. Differently from EVlncRNAs the The MOuse NOnCode Lung database (MONOCL) [197] includes information on mice infected with influenza virus and coronavirus causing severe acute respiratory syndrome (SARS-CoV). MONOCL is a database designed to retrieve annotations, expression profiles and functional enrichment results of lncRNAs expressed in lung. It has been demonstrated that lncRNAs impact on patient prognosis [198] and influence pathological conditions [114]. For these reasons, it is important to catalog lncRNAs linking their involvement with pathologies such as cancer. The Cancer LncRNA Census [199] is a catalog of lncRNA genes which have been causally implicated in cancer impact. Other databases that consider lncRNAs in association with cancer are the LncRNADisease database [200], the mammalian ncRNA disease repository [201]. For researchers that work with zebrafish in addition to the Ensebl database, it was developer the Zebrafish LncRNA Database (ZFLNC) [202]. Similarly, researchers that work with *S. scrofa* can use Ensembl or Pig lncRNA net [203]. CANTATA [204] is a database specific for lncRNAs expressed in plants.

Until now we described databases including gene loci and expression data about lncRNAs, but as previously discussed, to understand the function of a lncRNA it is important to know its subcellular localization. The database LncATLAS [205] displays the subcellular localization of lncRNAs annotated in GENCODE. This list of databases includes already working links; databases without working links were not included.

## 8. Conclusions and Perspectives

Non-coding RNAs appear to be the preponderant part of transcribed genomes in eukaryotes with a regulatory capacity in different physiopathological processes that were described in several manuscripts (see [206] for a review on miRNA function and [207,208] for lncRNAs). It would be interesting to understand their function in cell reprogramming considered results obtained by researchers studying reprogramming cells [156,209]. In fact, cell transformation requires the transformation of the cells with specific transcription factors. Unfortunately, this process has low efficiency and the original cocktail of reprogramming factors includes oncogenes that can lead to tumorigenesis. A step forward in the safety of cell reprogramming was done using miRNAs [210]. Characterizations of both miRNA and lncRNA reveal their interdependency to modulate cell reprogramming with lncRNAs that represent an additional regulatory layer in maintaining pluripotency and determining cell fate (reviewed in [211]). The analysis of lncRNAs requires a deeper mechanistic exploration of their cellular expression, subcellular localization, interacting partners to interpret their function in cell reprogramming and several other biological contexts. We and other researchers demonstrated the importance of using a single-cell approach to evidence the function of specific ncRNAs [113,114,212]. This result is related to the cell-specific expression of ncRNAs not detectable analyzing RNA expression using whole tissues or a heterogeneous cell population. For these reasons, recently developed techniques for the high throughput analysis of single cells (the first single-cell RNA-seq study was published in 2009 [110] and in 2011 the first single-cell DNA-seq [213]) may improve our knowledge in molecular events involved in maintaining complex interaction among same and different cell types within tissues. Moreover, these techniques may be useful for understanding alterations that trigger the development of several pathologies, such as tumors and clarify the molecular mechanisms of cell–cell interactions. To better elucidate the function of non-coding RNAs it would be useful the application of large-scale transcripts perturbation such as CRISPR/CAS9 system [214]. 

Single-cell RNA-seq techniques evidenced the possible existence of many more cell types than previously estimated [30], but a functional explanation of this phenomenon has not been provided yet. Two alternative hypotheses were. proposed: i) The “functionally neutrality” of alterations, and ii) an “adaptive” based mechanism. The first hypothesis is based on the possible existence of multiple equi-phenotypic molecular states (different combinations of gene expression may lead to the same phenotype) [215]. The second hypothesis is based on the evolutionary concept: Having cell-to-cell variability allows a better response to environmental changes [216]. If the single-cell sequencing gives us a photo of the molecular state for each analyzed cell it is limited to the number of analyzed molecules. For example, probably not all transcripts are recovered considering that from a single-cell analysis about 3000–4000 transcripts are detected while using a bulk sample the information increases to about 10,000 transcripts. The analysis of proteins using single cells is even more complicated due to the difficulty of protein amplification, which results in about 2000 analyzed proteins [217]. It would be interesting to understand if this discrepancy between the numbers obtained with single cells and bulk samples is due to a detection limit or if effectively single cells express such a limited number of genes and the union of the molecules expressed from different cells composing the tissue identifies the 10,000 activated genes that are found using bulk samples. Moreover, most single-cell expression protocols are based on RNA retrotranscription (RT) using oligod(T) as primers for the RT enzyme. This means that only RNAs having poly(A) have been retrotranscribed losing information on RNAs without poly(A) such as, for example, mature miRNAs, that are important non-coding RNAs involved in post-transcriptional gene regulation. Adaptor ligations to both 5′ and 3′ ends while using masking oligonucleotides to eliminate rRNAs may be an alternative method to evaluate the expression of non-polyadenylated RNAs in single cells avoiding sequencing of most abundant rRNAs in cells. This method was used by Faridani and colleagues [132] and allowed them to identify about 3800 miRNAs from HEK293, naïve embryonic stem cells, and primed embryonic stem cells. We used the SMART-seq based method to identify mature miRNAs expressed in skeletal muscle single fibers evidencing 137 miRNAs. We confirmed the expression of miRNAs detected with single-cell sequencing by using qRT-PCR showing an average correlation of almost 0.8 among the two techniques for 55 tested miRNAs [113]. As we profiled miRNAs and mRNAs from the same myofibers [113], also Xiao and colleagues [218] proposed the Holo-seq method to profile mRNAs and miRNAs from the same sample. As we obtained from single myofibers, they obtained the expression of about 250 miRNAs from single embryonic stem cells.

In this review, we considered the RNA expression, but single cells can be characterized not only at transcriptional but also at genomic [213,219] or proteomic [217,220] levels. It would be beneficial, for a better comprehension of single cell functions, to integrate all information such as, for example, genomic alterations that subtend to specific gene activation or repression and the related protein expression. Unfortunately, with the current technologies, all these data cannot be obtained from the same cell.

It is important to clarify that the analysis of single cells falls behind its importance if not associated with spatial information. Moreover, cells have their own identity because of the consistence (rigidity) of their ambient (extracellular matrix) [221,222,223]. The human body, for example, is formed by ~100 trillion cells each spatially located in specific tissues with different softness where exerting their functions in a collaborative network with other cells. To face such a complicated problem such as the spatial localization of cells from methodological, experimental, computational, and economic points of view it was launched a National Institutes of Health (NIH) initiative named Human Biomolecular Atlas Program (HuBMAP). It intends to develop a workflow for mapping the human body at single-cell resolution supporting technology development, data acquisition, and detailed spatial mapping [224]. A possible difficulty in these types of projects is determining the number of cells required to avoid losing information from very rare cells present in the tissue of interest. Data produced from this and related projects will be useful to formulate new biomedical hypotheses, to identify new biomarkers for disease diagnosis, to stage disease progression, and to predict optimal treatment regimens, to deeply comprising tissue functions and cell interactions for a step forward towards precision medicine. In this regard, here we largely discussed the importance of lncRNAs in developmental, pathological and physiological processes supporting the necessity of recovering their expression, cellular localization, and function at single-cell level. Obviously, the interpretation of such amount of data generated from single-cell analyses and spatial localization is challenging and need the development of specific algorithms. Standards are also needed to avoid the generation of inconsistent and not comparable data to allow to reproduce studies. Finally, the improvements of sequencing technologies, e.g., using better enzymes that limit errors and biases due to DNA amplification, will be beneficial for the development and diffusion of single-cell analysis. 

## Figures and Tables

**Figure 1 ijms-21-00302-f001:**
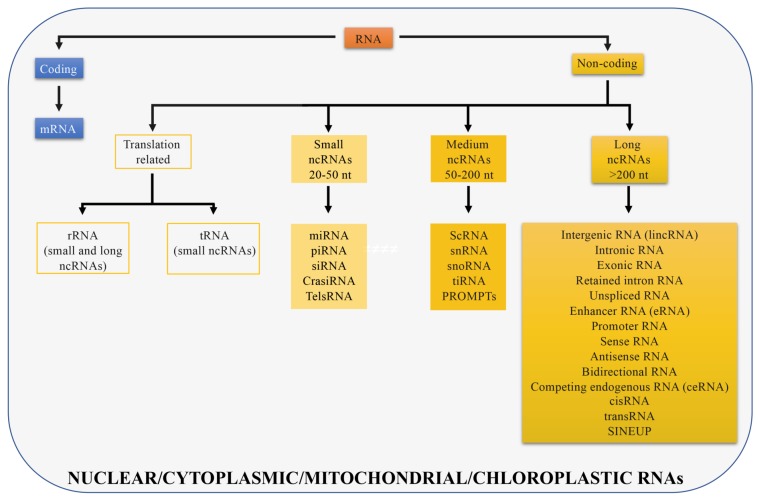
Different RNA types in cells. RNAs are divided in coding (if they are translated; blue) and non-coding (if they are not translated; yellow). Non-coding RNAs can be distinguished in different RNA types depending on their size or function. Long non-coding RNAs are longer than 200 nt and all displayed categories are discussed in the manuscript. Transcripts presented can be localized in the cell nucleus, cytoplasm, or in both compartments, in mitochondria or chloroplasts.

**Figure 2 ijms-21-00302-f002:**
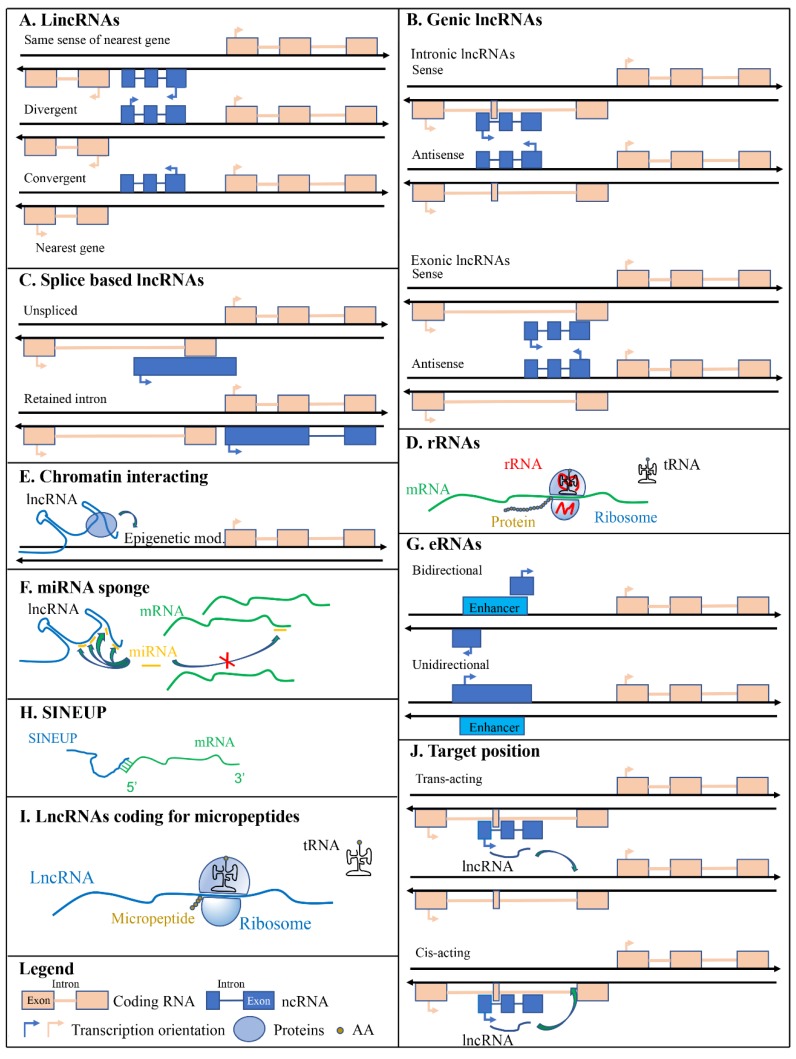
Scheme of different types of long non-coding RNAs (lncRNAs). (**A**) licRNAs are localized between coding genes and do not overlap any of them. (**B**) Genic lncRNAs are overlapped to coding sequences. (**C**) Splice variants. (**D**) Ribosomal RNAs are components of ribosomes. (**E**) Chromatin interacting lncRNAs allow chromatin modifications permitting the correct localization of modifying enzymes. (**F**) miRNA sponging lncRNAs reduce the availability of miRNAs to modulate the expression of their mRNA target. (**G**) Enhancer lncRNAs are transcribed from DNA regions with enhancing functions. (**H**) SINEUPs are lncRNAs that associating to mRNAs are able to enhance their translation. (**I**) lncRNAs can also code for micropeptides. (**J**) lncRNAs are distinguished according to the position of their targets. Green arrows indicate protein (**E**), miRNA (**F**), or lncRNA (**J**) targets. AA is for amino acids.

**Figure 3 ijms-21-00302-f003:**
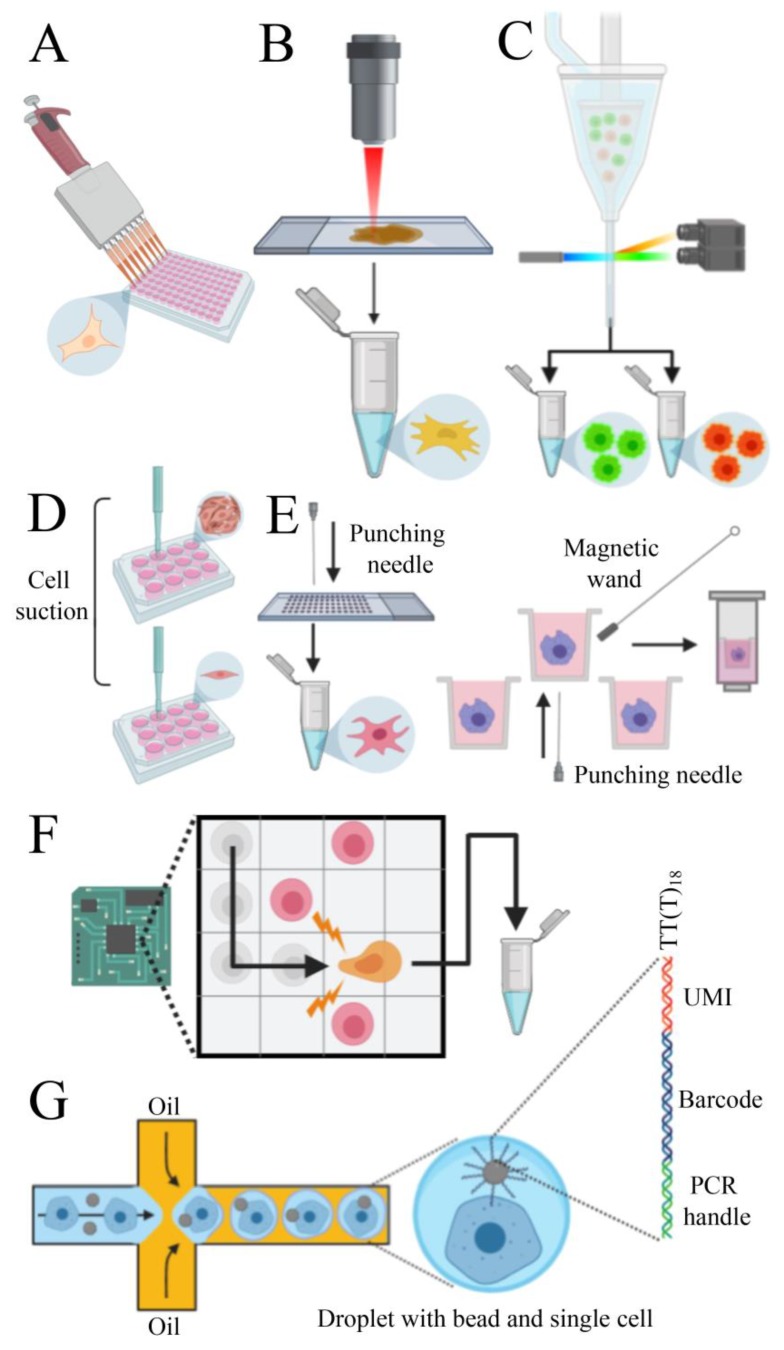
Methods to isolate single cells. (**A**) Micropipette isolation method consists of diluting the sample until obtaining a single cell on each well. (**B**) Laser capture microdissection utilizes a laser linked to a microscope to collect single cells from solid tissues. (**C**) Fluorescent activated cell sorting can sort cells tagged with fluorescent markers. (**D**) Capillary-based technology employs mechanical suction to isolate the cells. (**E**) Punching technologies are based on the principle of capturing cells on microwells and “punch” them on collection vials. (**F**) The dielectrophoresis system uses the electrokinetic properties of cells to move them on a chip. (**G**) Microfluidic platforms can capture single cells and barcoded beads into droplets of water in an oil phase. Each bead contains several primers composed by PCR handles, cell barcodes, and unique molecular identifiers (UMIs) sequences. PCR handles are common in all beads and allows PCR amplification after cDNA synthesis. Cell barcode sequences, identical across all the primers of the same microparticle but different from those on different beads, allow recovery of the cells’ origin. UMIs, different on each primer, allow mRNA transcripts to be digitally counted and to identify PCR duplicates. An oligod(T) sequence is present at the end of all primers for capturing polyadenylated RNAs and priming reverse transcription.

**Figure 4 ijms-21-00302-f004:**
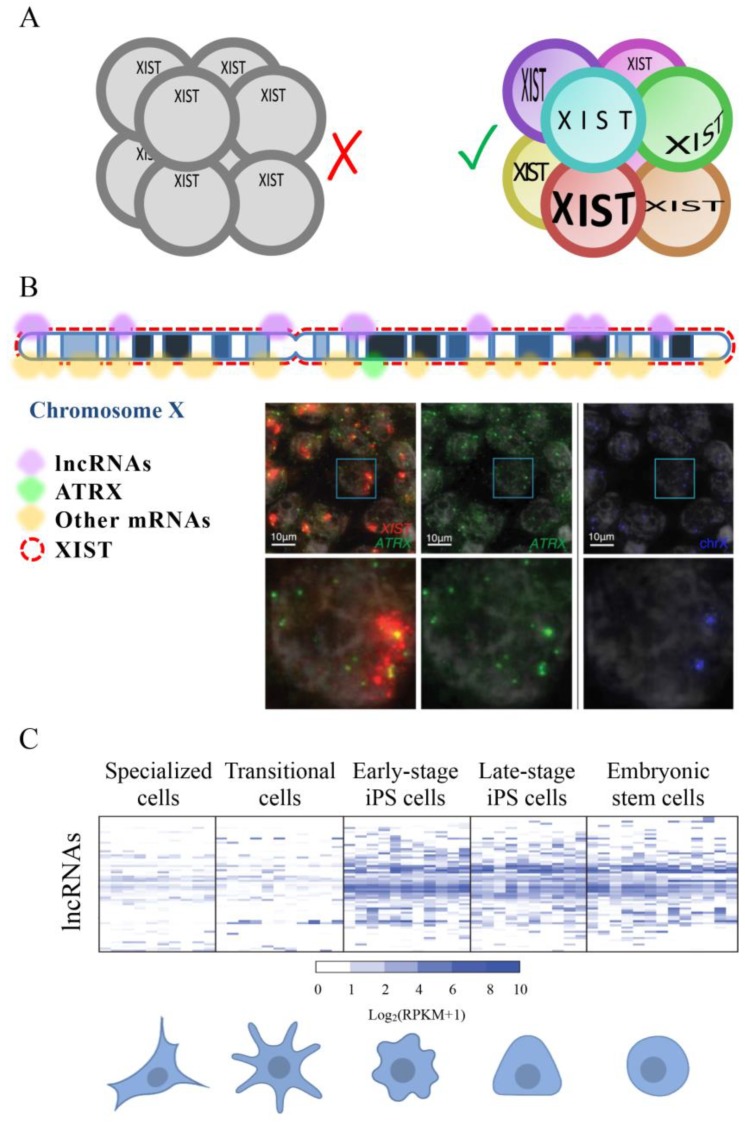
(**A**) Single-cell RNA sequencing of 8-cell embryos revealed that, contrary to what was previously believed, each cell has a different set of expressed genes and that the expression of XIST is not equal in the eight cells. (**B**) A scheme of an inactivated X chromosome with lncRNA XIST represented by red rectangles. Some genes are transcribed from this chromosome even if it is considered inactivated. Expressed lncRNAs are represented by purple semicircles while mRNAs are shown in yellow semicircles. In green, it is marked the position of ATRX, one of the mRNAs that escape the inactivation of the X chromosome. Microscope images show the expression of XIST (red), ATRX (green) and the position of the two X chromosomes (blue). Moreover, it is represented an enlargement of a single nucleus (white rectangle) for each staining. FISH images are modified from [155]. (**C**) Differences in lncRNA expression between cells at different stages of reprogramming. Some lncRNAs seem to be specifically expressed in the less specialized cells, suggesting that they might be involved with the maintenance of the pluripotency of stem cells and that might be used as tools to induce specialized cells to transition into less specialized cells (like iPSCs). Heat map is modified from [156].

**Table 1 ijms-21-00302-t001:** Comparison of different techniques used to isolate single cells. LCM: laser capture microdissection; FACS Fluorescent Activated Cell Sorting; DEP: dielectrophoresis.

Method	Micropipette Isolation	LCM	FACS	Capillary Based	Punching Technology	DEP	High-Throughput Droplet-Based	Low-Throughput Droplet-Based
Main Platforms	N/A	Several Platforms	Several Platforms	AVISO CellCelector	CellRaft AIR	DEPArray NxT	Chromium System	C1 System
							Nadia	
					Puncher Platform		InDrop System	
							ddSEQ Single-Cell Isolator	
**Throughput**	Low	Low	High	Low (<100 cells)	Low (<100 cells)	Low (<100 cells)	High (between 6 k and 10 k cells)	Low (<800 cells)
**Visual Control**	Yes	Yes	No	Yes	Yes	Yes	No	Yes
**Cell selection**	Yes (morphologically)	Yes	Yes	Yes	Yes	Yes	No	Yes
**Input of cells**	Low	Low	High	Medium	Medium	Medium	High	Medium
**Advantages**	Low cost	Spatial information, storage of tissue	Capture rare cells, fast analysis	Low price of consumables	Active cell selection, high transfer efficiency	Active cell selection, cell–cell interaction analysis	Suitable for processing a high number of cells	Suitable for RNA-Seq, DNA-Seq, miRNA-Seq, epigenomics, qRT-PCR analysis
**Disadvantages**	Laborious, low efficiency	Fixatives can damage RNA and introduce bias	Require antibodies/molecular markers	Require skills to operate, bioinformatics not provided	Bioinformatic analysis not provided	High price of consumables	High cost, profiles of only polyadenylated RNAs (need specific developed protocols for example miRNAs that are not polyadenylated)	Size-based cell selection

**Table 2 ijms-21-00302-t002:** List of databases storing single-cell data.

Database Name	Organisms	Reference
PanglaoDB	Mouse, human	[182]
Single-cell database	Mouse	[183]
JingleBells	Different organisms (Mouse, human, zebrafish, brown rat)	[184]
Brain atlas	Mouse, human	[185]
Single-cell RNA sequencing	Human	[186]
Single-cell data with Nadia (DolomiteBio)		[187]
Sanger institute experiments	Mouse, human	[188]
BioTuring		[189]
Cancer single-cell atlas	Human	[190]

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
