# Peer review of "A Single Cell but Many Different Transcripts: A Journey into the World of Long Non-Coding RNAs"

_ijms, 2020, doi:10.3390/ijms21010302_

Round 1

Reviewer 1 Report

This article reviews long non-coding RNAs (lncRNAs). It contains many materials such as classification of lncRNAs according to genomic position, function, etc. Databases for lncRNAs are also provided. However, it is not easy to read this paper.

Comments

It is not easy to read this paper. Perhaps, because it contains many materials, it does not provide an in-depth review or discussion on some subjects. Many of contents are only slightly mentioned and it does not provide useful information for readers. The article could be considerably shortened and revised to focus on some subjects. Line 79. Please explain ‘oligod(T)’. There are many places that you may need to explain, but not just mention it because readers may not be very familiar with them. The legend of Figure 1 is too long.

Author Response

We thank the Reviewers for their comments. We also modified some sentences making them more comprehensible. All changes in the manuscript are highlighted in red.

Reviewer 1

This article reviews long non-coding RNAs (lncRNAs). It contains many materials such as classification of lncRNAs according to genomic position, function, etc. Databases for lncRNAs are also provided. However, it is not easy to read this paper.

Comments

It is not easy to read this paper. Perhaps, because it contains many materials, it does not provide an in-depth review or discussion on some subjects. Many of contents are only slightly mentioned and it does not provide useful information for readers.

Authors. The review aims to reach a broad group of scientists that can find valuable information available on the literature summarized in a single manuscript. The manuscript is organized into four main sections (a. lncRNA classification, b. approaches for single-cell analysis, c. results regarding lncRNA functions and specificity obtained from the analysis of single cells, d. databases useful for the study of lncRNA). These sections are designed so that they can be read independently from each other to facilitate the reader in finding the information he/she is looking for.

We know that the decision of organizing the manuscript in such a way has increased its length, and the first part could be omitted, but we experienced that many colleagues could find useful to have a complete description of lncRNA classification and therefore we considered that is extremely important to include that part as well. We improved the abstract to provide a better explanation of the purpose of the review (lines 22-25).

The article could be considerably shortened and revised to focus on some subjects.

Authors. We thank the reviewer for his/her comment. As previously stated, one option to make the article shorter could be the removal of the section about lncRNA classification from the article, however we consider that type of review is interesting for scientists that are already working on the field of non-coding RNAs and for those that are approaching in just starting to approach the subject. Since this section was appreciated by Reviewer 2, we decided to not remove it from the article to avoid having to redirect readers to other articles to find information regarding this topic and to allow an overall better discussion about the topic of lncRNA in this review. 

Line 79. Please explain ‘oligod(T)’. There are many places that you may need to explain, but not just mention it because readers may not be very familiar with them.

Authors. We thank the Reviewer for this observation. We improved the text in order to avoid unnecessary confusion (lines 96-99).

The legend of Figure 1 is too long.

Authors. We thank the Reviewer for this observation. We shortened the length of the legend moving information in the main text (lines 55-75 and page 5 figure legend).

Reviewer 2 Report

In the review by Alessio et al, the authors have discussed the use of single-cell RNA sequencing for identification of long non-coding RNAs (lncRNAs). They have provided a detailed description of different classes of lncRNAs based on their genomic organization, functions and cellular localization. The authors have discussed the importance of single-cell RNA sequencing for identification of lncRNAs. They have also described different methods to obtain single cells for the single-cell RNA sequencing and highlighted the importance of one method over the others. Moreover, they have provided comprehensive discussion about the various library preparation methods to detect polyadenylated, non-polyadenylated RNAs and total RNAs. Finally, they provide a summary of studies where single-cell RNA-seq approaches have been employed for the identification of lncRNAs in embryo-derived cells, stem cells, differentiated cells and in tumors. In addition, they provide a list of available databases for the community to study lncRNAs in different species.

This review is nicely written and provides great details especially in the classification of noncoding RNA. It will be useful for the lncRNA community. However, I have some suggestions that can be incorporated to strengthen the review as follows:

In Figure 2, please number each panel as 2A, 2B, 2C and so on and refer each panel at the appropriate section in the text instead of just mentioning Figure 2. In Figure 2 (scheme of different types of lncRNA), the panels related to miRNA sponging and rRNAs are not clear. Figure legend might be useful.

Since, abundance of lncRNAs is usually lower as compared to coding genes, it would be helpful if authors can provide information mentioning the number of cells and sequencing depth required in order to identify lncRNAs from single-cell RNA-sequencing in different cell types.

It will be also useful to provide some information on how single-cell RNA-sequencing will determine lncRNA mechanisms in cells or tissues.

Abstract can be improved to clearly describe the purpose of the review.

Author Response

We thank the Reviewers for their comments. We also modified some sentences making them more comprehensible. All changes in the manuscript are highlighted in red.

Reviewer 2

In the review by Alessio et al, the authors have discussed the use of single-cell RNA sequencing for identification of long non-coding RNAs (lncRNAs). They have provided a detailed description of different classes of lncRNAs based on their genomic organization, functions and cellular localization. The authors have discussed the importance of single-cell RNA sequencing for identification of lncRNAs. They have also described different methods to obtain single cells for the single-cell RNA sequencing and highlighted the importance of one method over the others. Moreover, they have provided comprehensive discussion about the various library preparation methods to detect polyadenylated, non-polyadenylated RNAs and total RNAs. Finally, they provide a summary of studies where single-cell RNA-seq approaches have been employed for the identification of lncRNAs in embryo-derived cells, stem cells, differentiated cells and in tumors. In addition, they provide a list of available databases for the community to study lncRNAs in different species.

This review is nicely written and provides great details especially in the classification of noncoding RNA. It will be useful for the lncRNA community. However, I have some suggestions that can be incorporated to strengthen the review as follows:

In Figure 2, please number each panel as 2A, 2B, 2C and so on and refer each panel at the appropriate section in the text instead of just mentioning Figure 2. In Figure 2 (scheme of different types of lncRNA), the panels related to miRNA sponging and rRNAs are not clear. Figure legend might be useful.

Authors. We thank the Reviewer for this suggestion. We modified accordingly the figure 2 (also including lncRNAs coding for micropeptides 2I) and consequently the text (lines 124, 140, 159, 174, 179, 206, 214, 237, 244, 249, 265, 267) and figure legend (lines 280-287).

Since, abundance of lncRNAs is usually lower as compared to coding genes, it would be helpful if authors can provide information mentioning the number of cells and sequencing depth required in order to identify lncRNAs from single-cell RNA-sequencing in different cell types.

Authors. We agree with the Reviewer. We stressed the number of cells to analyze in single cell experiments (lines 427-429) and included information on the number of reads necessary for the analysis of lncRNAs (lines 482-490).

It will be also useful to provide some information on how single-cell RNA-sequencing will determine lncRNA mechanisms in cells or tissues.

Authors. We agree with the Reviewer that only performing single-cell RNA-sequencing is not sufficient to determine mechanisms of action of lncRNAs, but it permits to infer their function correlating their expression with one of the coding RNAs such as in Le Béguec et al (Sci Rep 2018) We included this explanation in the revised version of the manuscript (lines 685-692). Moreover, the analysis of single cells allows understanding the specificity of each transcript, as already stated at lines 492-493, for a better plan of silencing or activating experiments as we discussed in Alessio et al (NAR 2019). Moreover, as we discussed in the abstract, lncRNAs show a more cell-specific expression (lines 15-18) and their analysis at the single cell level allows to better discover their expression and cell specificity. We also included in the review the figure 4 to better explain the improvements that single cell analysis gave in understanding the function of specific lncRNAs. 

Abstract can be improved to clearly describe the purpose of the review.

Authors. We improved the abstract describing the purpose of the review (lines 22-25).

Round 2

Reviewer 1 Report

The authors have addressed my comments.